# Improved Outcome Prediction for Appendiceal Pseudomyxoma Peritonei by Integration of Cancer Cell and Stromal Transcriptional Profiles

**DOI:** 10.3390/cancers12061495

**Published:** 2020-06-08

**Authors:** Claudio Isella, Marco Vaira, Manuela Robella, Sara Erika Bellomo, Gabriele Picco, Alice Borsano, Andrea Mignone, Consalvo Petti, Roberta Porporato, Alexandra Ambra Ulla, Alberto Pisacane, Anna Sapino, Michele De Simone, Enzo Medico

**Affiliations:** 1Candiolo Cancer Institute—FPO, IRCCS, Strada Prov. 142, km 3.95, I-10060 Candiolo (TO), Italy; manuela.robella@ircc.it (M.R.); saraerika.bellomo@ircc.it (S.E.B.); gp9@sanger.ac.uk (G.P.); alice.borsano@ircc.it (A.B.); andrea.mignone@virgilio.it (A.M.); consalvo.petti@ircc.it (C.P.); roberta.porporato@ircc.it (R.P.); alexandra.ulla@ircc.it (A.A.U.); alberto.pisacane@ircc.it (A.P.); anna.sapino@ircc.it (A.S.); michele.desimone@ircc.it (M.D.S.); 2Department of Oncology, University of Torino, Strada Prov. 142, km 3,95, 10060 Candiolo (TO), Italy; 3Department of Medical Sciences, University of Torino, 10126 Torino, Italy

**Keywords:** appendiceal cancer, pseudomyxoma peritonei, peritoneal carcinosis, gene expression profiling, prognostic signatures, tumor stroma, cancer-associated fibroblasts.

## Abstract

In recent years, cytoreductive surgery (CRS) and hyperthermic intraperitoneal chemotherapy (HIPEC) have substantially improved the clinical outcome of pseudomyxoma peritonei (PMP) originating from mucinous appendiceal cancer. However, current histopathological grading of appendiceal PMP frequently fails in predicting disease outcome. We recently observed that the integration of cancer cell transcriptional traits with those of cancer-associated fibroblasts (CAFs) improves prognostic prediction for tumors of the large intestine. We therefore generated global expression profiles on a consecutive series of 24 PMP patients treated with CRS plus HIPEC. Multiple lesions were profiled for nine patients. We then used expression data to stratify the samples by a previously published “high-risk appendiceal cancer” (HRAC) signature and by a CAF signature that we previously developed for colorectal cancer, or by a combination of both. The prognostic value of the HRAC signature was confirmed in our cohort and further improved by integration of the CAF signature. Classification of cases profiled for multiple lesions revealed the existence of outlier samples and highlighted the need of profiling multiple PMP lesions to select representative samples for optimal performances. The integrated predictor was subsequently validated in an independent PMP cohort. These results provide new insights into PMP biology, revealing a previously unrecognized prognostic role of the stromal component and supporting integration of standard pathological grade with the HRAC and CAF transcriptional signatures to better predict disease outcome.

## 1. Introduction

The combination of cytoreductive surgery (CRS) and hyperthermic intraperitoneal chemotherapy (HIPEC) is nowadays recognized as the “gold standard” treatment for pseudomyxoma peritonei (PMP) originated from mucinous appendiceal tumors [1,2,3,4,5,6,7,8,9], due to favorable biologic behavior characterized by a pattern of late or non-invasive multifocal spread into tissues, with low risk of hematogenous dissemination. Although less frequently, PMP has also been described in association with mucinous tumors from other sites, including colon, ovary, gallbladder, pancreas, and urachus [10].

As shown in Table 1, CRS plus HIPEC has been extensively used to increase the five- and ten-year overall survival (OS) of patients with PMP of appendiceal origin [11,12,13,14,15,16,17,18,19]. However, disease-free survival (DFS) shows a significant and often unpredictable variability, with a non-negligible relapse rate in treated patients [20,21,22,23,24,25,26,27]. The relapse is sometimes re-treatable by CRS associated or not with HIPEC, but in many cases this is not feasible due to the carcinomatosis features, not amenable with a complete surgical cytoreduction [4].

Those patients are often treated by systemic chemotherapy with drugs that are commonly used on advanced colon cancer [28,29], with poor results not only for the recognized inefficacy on mucinous tumors, but also because, as reported by Levine [30,31], those two biological entities are probably different. To better understand which unexplored features or mechanisms may be responsible for biologic, and consequently prognostic, behavior of the disease, we explored different factors that are reported to have an impact both on disease relapse and overall survival [22,23,24,25,27]. The main currently used pathological classification systems for PMP include Ronnett’s [22] and World Health Organization (WHO) [25]. Apart from minor differences, both subdivide PMP in two major categories: high and low histological grade, with an “intermediate group” described in Ronnett’s classification. An additional consensus proposed by the Peritoneal Surface Oncology Group International (PSOGI) is available [32], but it is not extensively used in Italy nor other countries.

Notably, an extensive examination of multiple samples is needed to provide accurate classification, considering the huge amount of mucin filling the peritoneal cavity and the multiplicity of lesions [22,25].

Low-grade PMP is commonly reported to have a positive prognostic impact on OS and DFS [12,13]. Other factors described to improve prognosis of PMP are the completeness of cytoreduction (CC), less likely in the case of extensive involvement of the small bowel or its mesentery, center experience on CRS + HIPEC, and neoadjuvant systemic chemotherapy, while disease diffusion measured by the Peritoneal Carcinomatosis Index (PCI) and postoperative complications are negative prognostic factors [12,13,16,17,19]. Recent works have investigated the predictive and prognostic role of various blood-based markers, including neutrophil/lymphocyte, lymphocyte/monocyte, and platelet/lymphocyte ratios, in the field of colorectal cancer (CRC). As these measurements are inexpensive and easily calculated, they could have a good potential also for determining PMP prognosis [33].

In our experience, some patients show unexpected outcomes in relation to the histological classification. For instance, even if the distribution of the above-mentioned parameters was in line with those reported in the literature, some patients with low-grade PMP without any negative prognostic factors showed a bad prognosis. On the contrary, some high-grade PMP patients did not relapse, even in the presence of other negative prognostic features.

Indeed, availability of reliable molecular markers predicting PMP outcome is still and unmet clinical need, although promising results have been surfacing in recent years. Genomic analysis of PMP highlighted a transcriptional oncogenic signature, named “high-risk appendiceal cancer” (HRAC), that predicts PMP relapse [30,31]. In primary CRC, we have shown that a transcriptional signature reflecting cancer-associated fibroblast (CAF) abundance and function (the “CAF-score”) has strong prognostic value [34,35,36]. It is therefore conceivable that the CAF signature, either alone or in combination with the HRAC score, provides additional prognostic information also for PMP. To characterize the molecular heterogeneity and biological variability of PMP, and to evaluate the prognostic performances of the HRAC signature of the CAF-score and of their integration, we performed global gene expression profiling on a collection PMP cases with full clinical annotation and long-term follow-up.

The analysis is particularly valuable considering the homogeneity of treatment- and institution-related variables, including the surgical and pathology teams. Indeed, the same team performed all surgical procedures using the same technique, temperature, and drugs, and dedicated pathologists performed multiple and extensive sampling of the disease to minimize the misdiagnosis risk for this heterogeneous disease.

## 2. Results

### 2.1. PMP Collection

From August 1997 to December 2018, 177 patients diagnosed with PMP underwent CRS + HIPEC. For clinical features of patients see Appendix A. The patients were treated by CRS with the technique described by Sugarbaker [15] and by HIPEC with cis-platinum (CDDP) 100 mg/m^2^ + C-mytomicin (MMC) 16 mg/m^2^ for 60 min, using an original semiclosed technique published by our team in 2003 [37]. Ronnett’s histopathological classification [22], which became available since we started to perform CRS + HIPEC, and is still employed today, was implemented in 2010 by WHO classification. Histopathologic classification of the patients is summarized in Appendix A. The DFS and OS of the cohort were 63.5 months and 126 months, respectively.

For the present work, tissue specimens from 35 consecutive patients of PMP treated by CRS + HIPEC between January 2014 and December 2016 in our institution were collected and snap-frozen for gene expression profiling with DNA microarrays. A control group consisted of surgical samples from 11 CRC peritoneal carcinomatosis, also collected and profiled. From the initial number of 35 PMP cases, 11 were excluded from subsequent analyses due to their histopathology (signet ring cells, neuroendocrine tumors, ovarian PMP) or because they were lost at follow-up. Of the remaining 24 patients, 13 had low-grade PMP histopathology, also named “diffuse peritoneal adenomucinosis” (DPAM), while 11 had high-grade or intermediate disease, also named “peritoneal mucinous carcinomatosis” (PMCA). All patients underwent complete CRS (CC-0/1) and HIPEC and had a follow-up of at least 24 months. Histopathological and clinical features of this subset of patients are provided in Appendix A.

### 2.2. Prognostic Value of the HRAC Signature in Multi-Lesion Profiles of PMP

Previous reports have identified an HRAC signature of 139 genes, predicting PMP recurrence [30,31]. Of note, in previous works only one lesion was profiled for each patient, although a distinctive property PMP is locoregional spreading in the peritoneum, leading to multifocal and potentially heterogeneous disease. PMP is a rare disease; therefore, validation of the HRAC signature in cohorts from independent centers was still missing, together with multiple lesion profile analyses. To tackle these issues, we generated transcriptional profiles from multiple lesions for 9 patients to evaluate heterogeneity in prognostic scoring (Figure 1a).

We calculated for each sample the HRAC score (Appendix A) and found that four of the nine patients with multiple profiles displayed a wide score range, resulting in both positive and negative values, therefore predicting a different outcome for different lesions of the same patient (Figure 1b). This observation is in line with standard histopathological practice, requiring evaluation of multiple PMP lesions for reliable classification [22,25]. Furthermore, it indicates that the HRAC score is not always consistent across different lesions of the same disease, possibly leading to unreliable classification.

To further explore this issue, we performed receiver operating characteristic (ROC) analysis on DFS censored at 24 months (Figure 1c). We observed that, despite the apparent heterogeneity, the prediction value of HRAC score was confirmed in the multi-sample dataset, both in the full collection, with an area under the curve (AUC) of 0.665, as well as separately in the DPAM (AUC = 0.704) and PMCA (AUC = 0.727) subgroups. Interestingly, when the global transcriptome profile of the multiple samples was used to select the most representative “central” sample, that is, the one closest to the averaged global gene expression of all samples for that patient, the prognostic performance was substantially improved (Figure 1c). Other strategies for multi-sample analysis, based on averaging the samples or the scores, or on considering the highest/middle/lowest score, had weaker association with prognosis. These results indicate that multi-sample profiling and selection of a central, representative sample is a suitable strategy to improve the prognostic performance of the HRAC signature.

Overall, this analysis independently confirmed the prognostic value of the HRAC signature; however, it also highlighted that the multifocality and heterogeneity of PMP could lead to suboptimal molecular classification, unless multiple lesions are profiled.

### 2.3. Prognostic Value for PMP of the Stromal CAF Score

It is widely recognized that stromal cells are important components of the tumor microenvironment, sustaining growth, progression, and in some cases therapy resistance. Previously, we and others found that transcriptional signatures for specific stromal components reflect CRC biology, and that a CAF gene expression signature predicts prognosis and response to therapy [34,35,38]. To evaluate a possible prognostic value also in PMP, we calculated the CAF score on an extended dataset also including 11 CRC samples (Figure 2a and Appendix A). Notably, PMP samples displayed systematically higher CAF scores (*t*-test *p*-value < 1 × 10^−7^), suggesting the prevalence of such stromal cells in the disease as previously mentioned [39]. This observation supports the notion that, despite not being routinely characterized in histopathological analysis, stromal cells—in particular CAFs—may be involved in PMP onset and progression. However, as for HRAC, also the CAF score in some cases was widely variable in multiple samples from the same patient (Figure 2b). This could be attributed to different tumor/stroma contents in different lesions as well as to bias in tumor sampling [36]. To verify whether the CAF score could provide information on disease outcome, we performed ROC analysis (Figure 2c) and found that the score predicts prognosis in the DPAM subgroup only after selecting the central samples (AUC = 0.8). No significance could be achieved in the PMCA subgroup and in the full cohort. Taken together, these results support the involvement of stromal cells, and in particular of CAFs, in PMP biology and suggest their involvement in the definition of a subgroup of patients with poor outcome.

### 2.4. Integrating CAF and HRAC Scores

Interestingly, both CAF and HRAC scores provided information on disease outcome, in particular in DPAM patients. Nevertheless, they were not significantly correlated to each other (*R*^2^ = 0.01). In addition, the overlap between the classifications was partial, as shown in Figure 3a. Most importantly, patients classified as low risk (negative value) by both scores had a much better outcome with long-term DFS, while patients called as high risk (positive value) by any of the two scores suffered from early recurrence. These data suggest that the CAF and HRAC scores provide different information on PMP biology and outcome.

As a logical consequence, we conceived an integrated algorithm in which positivity to either score is indicative of poor prognosis. The integrative classifier successfully stratified prognosis in the DPAM cohort (log-rank test = 7.36, *p* < 0.01). This classification was more effective than HRAC alone, which could only achieve marginal significance (log-rank test = 3.7, *p* = 0.054; Figure 3b), likely due to the limited size of the dataset.

To validate these observations, we took advantage of a public collection of PMP gene expression profiles (GSE75535), including 39 patients with single lesion profiles [30]. This dataset was originally employed for the validation of the HRAC signature, so we calculated the CAF score as previously described [34] to evaluate the prognostic value of the combined classifier (Appendix A). Accordingly, also in this independent cohort, the original HRAC classification (log rank *p* < 0.01) was further improved by combination with the CAF score (log-rank test = 7.73 *p* < 0.005), supporting the integration of the two scores (Figure 3c).

## 3. Discussion

Global gene expression analysis has been successfully applied in translational cancer research, providing new insight into biology and paving the way to the application of new solutions to predict disease outcome and response to therapies [35,40,41,42,43]. Typically, such analysis provides an aggregate profile of the cellular components that make up the tumor bulk, including cancer cells and stromal cells. Indeed, we and others have observed that stromal contribution to tumor gene expression is particularly relevant and, when characterized, allows a new level of stratification with informative prognostic value [34,35,36,44,45].

Typically, prognostic studies are applied to early-stage tumors in a clinical scenario in which the tumor outcome after surgery is uncertain. These conditions are usually met before detectable dissemination of the disease to other anatomical sites occurs. In most PMP patients, however, despite a high level of dissemination within the peritoneum, the disease is indolent, especially in the case of low-grade DPAM. CRS + HIPEC surgery treatment grants a good disease-free survival after surgery and long-term survival. Yet, several patients undergoing unpredicted relapse might benefit from different follow-up and treatment [20].

Previous work on PMP successfully identified the HRAC transcriptional signature as a predictor of disease outcome; however, these works did not capture the typical heterogeneity and multifocal nature of this disease [16,30]. Here, we successfully classified tumors according to the HRAC score and evaluated the intra-patient score variability. Interestingly such scores can be quite variable indicating that the HRAC score is not consistent across different lesions of the same patient. Indeed, the heterogeneity of PMP cannot be underestimated, and routine pathological analysis of the disease includes the characterization of more than one lesion [22,25]. Accordingly, the HRAC score was associated with DFS to a certain extent in our series but was further improved when the most representative sample for each patient was selected. This observation suggests that future analyses aimed at bringing transcriptional classification of PMP towards clinical application will have to consider profiling multiple lesions to properly capture disease aggressiveness.

Of note, the CAF score was similarly affected by scoring heterogeneity. Consequently, also in this case, multiple sampling greatly improved classification performance. This likely is due to the fact that, together with the biological variability of the tumor biology in different lesions, we had to superimpose the variability of the sampling of stromal cells, as previously mentioned [34].

Interestingly, the HRAC and CAF scores displayed only partial overlap in patient stratification, suggesting that these signatures highlight different aspects of PMP biology and support potentially different perspectives of clinical intervention. Most importantly, here we show that the CAF and HRAC scores can be efficiently integrated obtain superior prognostic performance.

From the biological and therapeutic standpoints, functional characterization of tumor samples harboring high CAF or HRAC scores could provide insight in the tissue of origin and functions of circulating microRNAs recently reported to be diagnostic of PMP [46]. The scores could also be investigated in light of reports showing association of mutations of the *KRAS* and *GNAS* genes with worse progression-free survival (PFS) [47,48,49,50,51]. These somatic mutations support treatment with capecitabine and bevacizumab, and in the case of KRAS (G12C) mutation also with novel, tailored strategies [52,53]. In this way, transcriptional prognostic classification could distinguish the patients with worse prognosis, and mutational analysis could corroborate the diagnosis and suggest optimal adjuvant therapy.

A potential limitation of our study is the limited sample size of the cohort of patients. This is a common issue in the field, since the disease is rare and its outcome is relatively good, so the collection of informative follow-up requires long observation times. This problem is shared also by public datasets that are further limited by the lack of multi-lesion profiles, which may lead to underestimation of the CAF score performance. In any case, these data show that exploring the contribution of the stromal populations can lead to improvements in the stratification of PMP patients, prompting further investigations on a larger scale together with the revision of the tumor specimen histology considering stromal cells for clinical practice.

## 4. Materials and Methods

### 4.1. Sample Collection

A total of 83 samples were obtained from patients who had undergone surgical resection at the Candiolo Cancer Institute. All patients provided informed consent, and study approval (protocol “Profiling” 001-IRCC-00IIS-10) was obtained from the review boards of the three institutions (the ‘Comitato Etico Istituto di Candiolo—FPO IRCCS’, the ‘Comitato Etico Azienda Ospedaliera Mauriziano Umberto I’ and the ‘Comitato Etico Azienda Ospedaliero-Universitaria Citta’ della Salute e della Scienza’). Immediately after resection, tumor specimens were macroscopically processed for histological analysis by a pathologist and sampled for snap freezing in liquid nitrogen. Frozen samples were maintained at −80 °C. Tissue analysis, qualification, and processing were conducted using standard procedures. Tumor tissue content was verified on frozen sections and the coupled diagnostic sections of each case were reviewed by two pathologists. Tumors were divided into two groups (high grade and low grade), following the Ronnett and WHO systems [22,25]. The peritoneal lesions were used for grading. Tumors that demonstrated any of the following were classified as high grade: high-grade nuclear atypia to include prominent nucleoli, hyperchromatic nuclei, markedly irregular nuclear membrane or dense coarse chromatin; architectural complexity, such as cribriform pattern, papillary formation, or extensive nuclear stratification; high cellularity; frequent mitosis; and signet ring cells.

### 4.2. Microarray Data Generation and Data Analysis

RNA was extracted using the miRNeasy Mini Kit (Qiagen, Hilden, Germany), according to the manufacturer’s protocol. Synthesis of cDNA and biotinylated cRNA (from 500 ng total RNA) was performed using the IlluminaTotalPrep RNA Amplification Kit (Ambion, Austin, TX, USA), according to the manufacturer’s protocol. Quality assessment and quantitation of total RNA and cRNAs were performed with Agilent RNA kits on a Bioanalyzer 2100 (Agilent, Santa Clara, CA, USA). Hybridization of cRNAs (750 ng) was carried out using Illumina Human 48 k gene chips (Human HT-12 V4 BeadChip, Illumina, San Diego, CA, USA). Array washing was performed by Illumina High Temp Wash Buffer (Illumina, San Diego, CA, USA) for 10 min at 55 °C, followed by staining using streptavidin-Cy3 dyes (Amersham Biosciences, Little Chalfont, UK). Hybridized arrays were stained and scanned in a Beadstation 500 (Illumina, San Diego, CA, USA). Expression data were deposited in Gene expression omnibus (GEO: https://www.ncbi.nlm.nih.gov/geo/, ID: GSE147762). Probe intensity data were extracted using the Illumina Genome Studio software (Genome Studio V2011.1) and normalized with Loess normalization using the Lumi R package [54,55]. Probes were filtered to select those that showed detectable signal (detection *p* value = 0) in at least one sample. When multiple probes targeted the same gene, the probe with the highest signal variance was selected. The HRAC score was calculated as the average expression of signature genes [30] in log2 ratio space versus the central sample (the central sample is the one most correlated with the averaged gene expression profile of the cohort); to assign each sample to the low risk or high risk group, the median score calculated in the cohort was employed as threshold. CAF score was calculated as previously described [34] as the average expression of CAF genes in the linear signal space.

### 4.3. Survival Analyses

Overall survival (OS) time was computed as the date of CRS + HIPEC to the last known date of clinical follow-up (censored) or death resulting from progressive disease (event). DFS time was computed as the date of CRS/HIPEC to the first subsequent clinical follow-up resulting in diagnosis of clinically evident disease (event), or the last known date of clinical follow-up without clinical evidence of disease (censored). The Kaplan–Meier survival analysis was used to estimate patient and DFS survival rates. Log-rank *p* values < 0.05 were considered statistically significant. Cox proportional hazards regression was used to assess the prognostic significance of the gene-based classification system in the presence of conventional clinical variables. Significance for ROC curves was evaluated with the De Long test.

## 5. Conclusions

Correct outcome prediction is paramount for optimal treatment and follow-up of patients with peritoneal carcinosis from appendiceal cancer. This study shows that transcriptional profiles of PMP can provide improved prognostic stratification when two gene expression signatures are integrated: the HRAC signature, capturing cancer cell aggressiveness, and the CAF signature, capturing stromal contribution to PMP biology. Most importantly, both signatures work best when multiple lesions are sampled from the same patient and the central one is used for classification. These findings highlight the importance of considering the multifocal and heterogeneous nature of PMP for molecular stratification and clinical management.

## Figures and Tables

**Figure 1 cancers-12-01495-f001:**
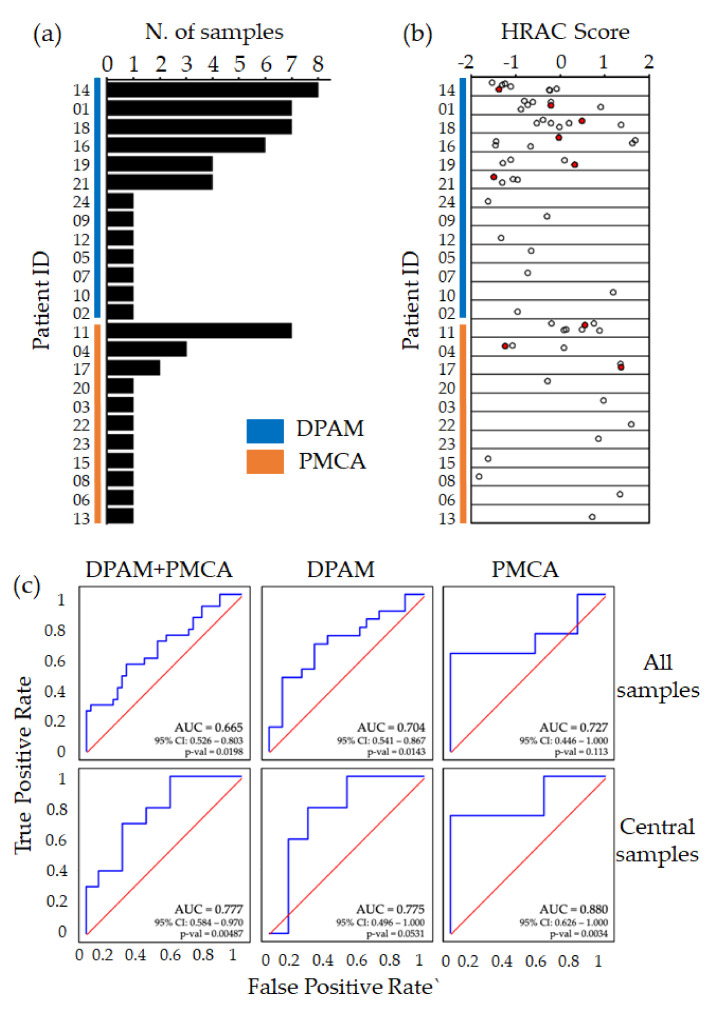
Prognostic performance of the transcriptional HRAC score. (**a**) Bar plot representing the cohort of 24 pseudomyxoma peritonei (PMP) patients in terms of number of samples per patient and pathological classification, with diffuse peritoneal adenomucinosis (DPAM) in blue and peritoneal mucinous carcinomatosis (PMCA) in orange; (**b**) high-risk appendiceal cancer (HRAC) score calculated in each sample subdivided by patient of origin. Red dots represent the central sample for patients with multiple profiles; (**c**) receiver operating characteristic (ROC) curves on disease-free survival (DFS) for HRAC in the full dataset (DPAM + PMCA), DPAM, and PMCA (DFS event = relapse within 24 months). For patients with multiple profiles, curves in the top row include all samples as if they were from different patients, while curves in the bottom row only include central samples.

**Figure 2 cancers-12-01495-f002:**
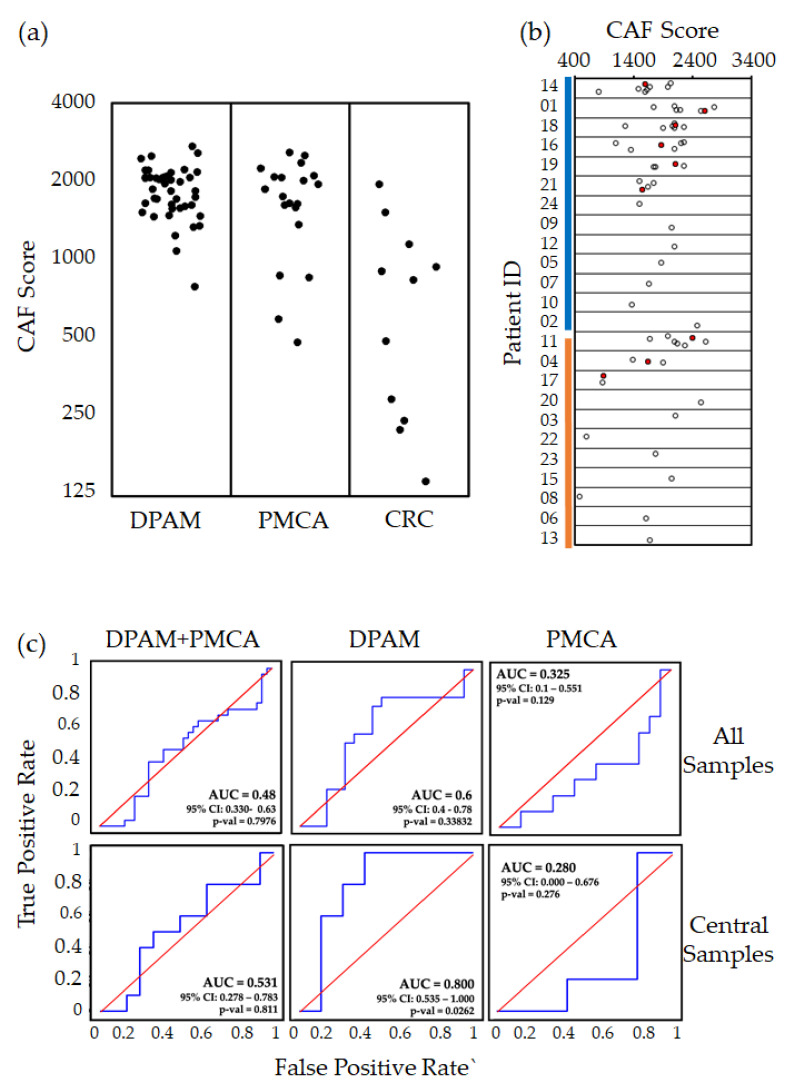
Prognostic performance of the transcriptional CAF score (**a**) Dot plot of cancer-associated fibroblast (CAF) score in DPAM, PMCA, and colorectal cancer (CRC) samples. (**b**) CAF score calculated in each sample subdivided by patient of origin. Red dots represent the central sample for patients with multiple profiles. (**c**) ROC curves for the CAF score vs DFS in the full dataset (DPAM + PMCA), DPAM, and PMCA (DFS event = relapse within 24 months). For patients with multiple profiles, curves in the top row include all samples as if they were from different patients, while curves in the bottom row only include central samples. AUC = area under the curve; CI = confidence interval.

**Figure 3 cancers-12-01495-f003:**
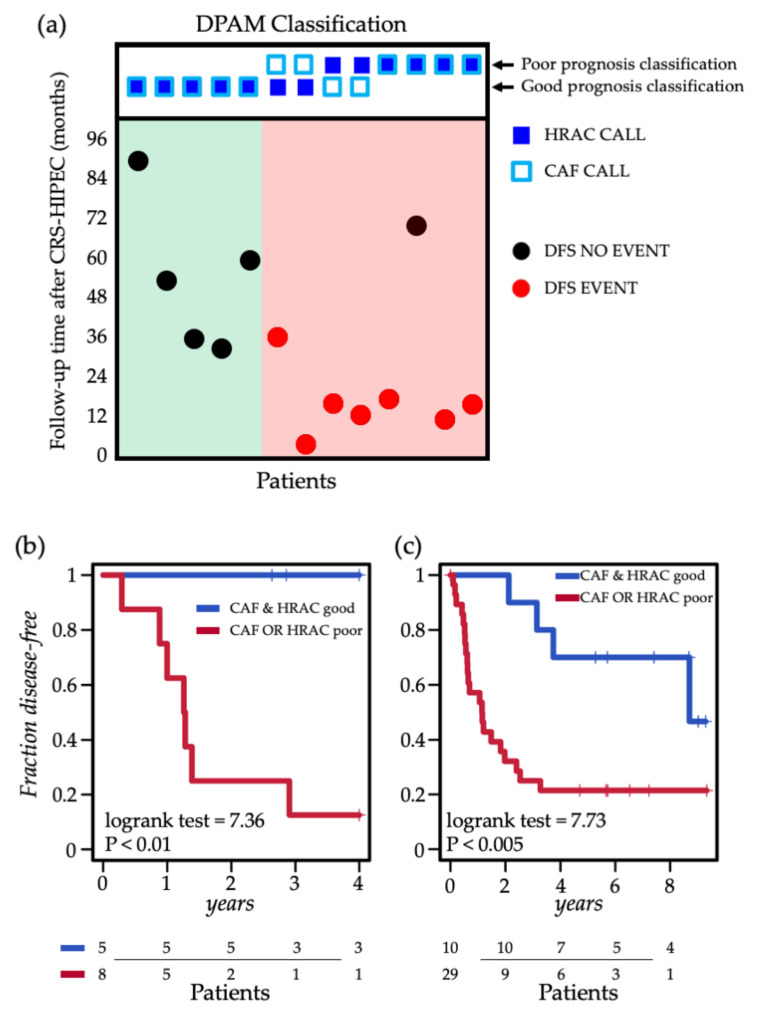
Combined prognostic performance of the HRAC and CAF scores. (**a**) Classification of the 13 DPAM cases according to CAF and HRAC scores on central samples with DFS annotation (y axis); samples are subdivided on the basis of HRAC and CAF calls (upper panel), and subdivided in 2 groups: CAF AND HRAC good (green background) and CAF OR HRAC poor (red background). (**b**) Kaplan–Meier plot of DFS of HRAC and CAF score integration in the 13 DPAM cases. (**c**) Kaplan–Meier plot of DFS of HRAC and CAF score integration in the GSE75535 dataset.

**Table 1 cancers-12-01495-t001:** Clinical outcome of appendiceal PMP treated with CRS + HIPEC.

References	DFS	5 Years OS	10 Years OS
Sugarbaker 2001 [15]	–	86%	62%
Mukherjee 2004 [14]	–	74%	–
Smeenk 2007 [9]	40 months	75%	–
Yan 2007 [19]	–	96%	–
Smeenk 2008 [2]	–	–	63%
Moran 2008 [18]	–	72%	55%
Chua 2012 [16]	98 months	74%	63%
Koh 2013 [6]	–	90%	–
McBride 2013 [17]	–	79%	55.9%
Taflampas 2014 [7]	66 months	91%	82%

OS: overall survival; DFS: disease-free survival; PMP: pseudomyxoma peritonei; CRS: cytoreductive surgery; HIPEC: hyperthermic intraperitoneal chemotherapy.

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
