# Peer review of "Improved Outcome Prediction for Appendiceal Pseudomyxoma Peritonei by Integration of Cancer Cell and Stromal Transcriptional Profiles"

_cancers, 2020, doi:10.3390/cancers12061495_

Round 1

Reviewer 1 Report

I read with great interest and appreciation the manuscript entitled "Improved outcome prediction for appendiceal pseudomyxoma peritonei by integration of cancer cell and stromal transcriptional profiles"  this work builds logically off previous publications in the literature to define molecular subtypes of a heterogeneous clinical condition of pseudomyxoma peritonei.  One of the most important findings and something that is congruent with what we see clinically is that there is significant tumor heterogeneity and that robust tissue sampling is critical to obtaining an accurate molecular profile whether it be the tumor itself or the cancer-associated fibroblasts (CAFs).  The unique contribution that the authors make to what Dr. Levine and the Wake Forest team have previously demonstrated is the integration of the CAFs into the prognostication.  The authors clearly demonstrate that the "high-risk appendiceal cancer" HRAC molecular signature was further improved by integration of the CAF signature.  Overall I think the manuscript was well written and will be of significant interest to the readership and add meaningfully to the current literature.  My only suggestions for improvement are below

1) In the Introduction line 48-49 on page 2 I think the authors need to add that the tumors are (biologically) different

2) Introduction pate 2 line 51-54 I think the authors should include citation and discussion of the PSOGI consensus paper Carr NJ, et al.  Am J Surg Path 2016 Jan 40(1) 14-26.

3) In the results section on page 2.  A few points of clarification.  Were all patients treated with mitomycin C hipec?  Also the authors describe the overall cohort of 177 patients over 21 years.  Over what time period were the 35 consecutive patients who were included in the study obtained. 

4) Results section page 3 last paragraph.  If I understand the "central" sample concept do multiple lesions have to be profiled?  Or does the pathologist just need to review multiple lesions to select one representative lesion to be profiled?  Could the authors clarify .

5) Maybe I missed in in Figure 3. but can the authors use either a table, figure or text to explain or illustrate the breakdown of DPAM and PMCA into the HRAC and CAF +/- cohorts. 

6) In the methods the authors state that overall survival was censored for death from progressive disease.  This would then be disease-specific survival.  This is usually challenging as some deaths are due to unknown causes.  How did the authors handle this?

Reviewer 2 Report

In this article Isella C et al, provide insight into pseudomyxoma peritonei biology, revealing a previously unrecognized prognostic role of the stromal component and supporting integration of standard pathological grade with the high-risk appendiceal cancer and cancer-associated fibroblasts transcriptional signatures to better predict disease outcome. The paper definitely deserves to be published and is a valuable contribution to the “cancers”. Some minor comments before publication.

Minor points:

[1] “Introduction”, First paragraph:

Pseudomyxoma peritonei refers to the accumulation of mucin within the peritoneal cavity secondary to mucinous appendiceal tumor. However, pseudomyxoma peritonei has also been described with mucinous tumors from other sites, including colon, ovary, gallbladder, pancreas, and urachus.

Recommended reference: Misdraji J. Mucinous epithelial neoplasms of the appendix and pseudomyxoma peritonei. Mod Pathol. 2015 Jan;28 Suppl 1:S67-79.

[2] “Introduction”, First paragraph:

The paper could be substantially improved with the incorporation of a table summarizing the most important data, including 5- and 10-year survival rates from references 1-18. This is key element for an article to be attractive for the readers.

[3] “Introduction”, Lines 57-76:

Many studies have investigated the predictive and prognostic role of various blood based inflammatory markers, including neutrophil–lymphocyte ratio (NLR), lymphocyte–monocyte ratio (LMR), and platelet–lymphocyte ratio (PLR) in the field of colorectal cancer. As NLR is both inexpensive and easily calculated, it has strong potential to be used in determining prognosis for patients with pseudomyxoma peritonei.

Recommended reference: Boussios S, et al. The Developing Story of Predictive Biomarkers in Colorectal Cancer. J Pers Med. 2019;9(1).

[4] “Introduction”, Line 58:

“Other factors described to improve prognosis of PMP are the completeness of cytoreduction…”.

Please, report that extensive involvement of the small bowel or mesentery predicts a high likelihood of incomplete cytoreduction.

[5] “Discussion”:

Please, make a comment about the potential role of circulating serum microRNAs as diagnostic biomarkers for pseudomyxoma peritonei.

Recommended reference: Song Y, et al. Circulating serum microRNAs as diagnostic biomarkers for pseudomyxoma peritonei. Int J Clin Exp Med 2018;11(11):12340-12346.

[6] “Discussion”:

What about the KRAS and GNAS mutations in pseudomyxoma peritonei? It has been shown to have negative prognostic value.

Recommended reference: Pietrantonio F, et al. Toward the molecular dissection of peritoneal pseudomyxoma. Ann Oncol. 2016 Nov;27(11):2097-2103.

[7] “Discussion”:

Metronomic capecitabine and bevacizumab is an active and well tolerated option in patients with relapsed pseudomyxoma peritonei.

Recommended reference: Pietrantonio F, et al. GNAS mutations as prognostic biomarker in patients with relapsed peritoneal pseudomyxoma receiving metronomic capecitabine and bevacizumab: a clinical and translational study. J Transl Med. 2016 May 6;14(1):125.
